



# Flow dependence of wintertime subseasonal prediction skill over Europe

Constantin Ardilouze[1], Damien Specq[1], Lauriane Batté[1], and Christophe Cassou[2]

[1]CNRM, Université de Toulouse, Météo-France, CNRS, Toulouse, France
[2]CECI, Université de Toulouse, CNRS, CERFACS, Toulouse, France

**Correspondence:** Constantin Ardilouze (constantin.ardilouze@meteo.fr)

**Abstract.** Issuing skillful forecasts beyond the typical horizon of weather predictability remains a challenge actively addressed by the scientific community. This study evaluates winter subseasonal reforecasts delivered by the CNRM and ECMWF dynamical systems and identifies that the level of skill for predicting temperature in Europe varies fairly consistently in both systems. In particular, forecasts initialized during positive NAO phases tend to be more skillful over Europe at week 3 in both systems.

Composite analyses performed in an atmospheric reanalysis, a long-term climate simulation and both forecast systems unveil very similar temperature and sea-level pressure patterns three weeks after NAO conditions. Furthermore, regressing these fields onto the 3-week prior NAO index in a reanalysis shows consistent patterns over Europe but also other regions of the northern hemisphere extratropics, thereby suggesting a lagged teleconnection, either related to the persistence or recurrence of the postive and negative phases of the NAO. This teleconnection, conditionned to the intensity of the initial NAO phase, is well

captured by forecast systems. As a result, it is a key mechanism for determining a priori confidence in the skill of wintertime subseasonal forecasts over Europe as well as others parts of the northern hemisphere.

## 1   Introduction

Skillful weather and climate predictions for horizons beyond two weeks could benefit many users (White et al., 2017). Lately, so-called subseasonal-to-seasonal (S2S) forecasts have gained considerable attention and effort from the scientific community

in multiple aspects, including the characterization of sources of predictability, such as atmospheric teleconnections, the initialization and generation of ensemble forecasts, the calibration and tailoring of the raw model outputs for enhanced usability and uptake by the application community (Merryfield et al., 2020). The S2S horizon has been often considered as a "predictability well" based on mean statistics and traditionnal methods and analyses inspired from seasonal-to-decadal climate prediction, but the most recent studies reveals instead so-called windows of opportunity based on the fact that under certain circumstances,

and for specific events and regions, S2S predictability can be considerably increased (Mariotti et al., 2020). This conditional predictability is illustrated by a number of case studies showing the successful anticipation of extreme climate events by dynamical forecast systems beyond 15-day lead time (Domeisen et al., In revision). The a priori identification of these windows of opportunity is a major asset in an operational context for the development and uptake of climate services relying on subseasonal forecasts, but it remains a scientific challenge with some promising examples. For instance, Mayer and Barnes (2021) recently





related the accuracy of North-Atlantic geopotential height forecasts issued by a neural network to their level of confidence. Their approach also allows to pinpoint the most relevant remote tropical regions leading to higher forecast skill.

At subseasonal scale, European and Eastern North American climate are influenced by the phase of the North Atlantic Oscillation (NAO), the leading mode of climate variability over the North Atlantic sector (Cattiaux et al., 2010; Seager et al., 2010; Luo et al., 2020). The positive (NAO+) and negative (NAO-) phases of the NAO correspond to two well-identified weather regimes characterizing recurrent synoptic-scale atmospheric patterns in winter, along with the Atlantic Ridge (AR) and Scandinavian Blocking (BLO) (e.g. Vautard, 1990). The NAO is sometimes considered as the local manifestation of a hemispheric variability pattern called Northern Annular Mode or Arctic Oscillation (AO). AO and NAO are strongly correlated in present climate (Hamouda et al., 2021).

The tropics-extratropics teleconnection described by Cassou (2008) illustrates the major role of the Madden-Julian Oscillation (MJO) phase in pre-conditioning North Atlantic weather regimes. Recently, Lee et al. (2019) found evidence of El-Niño Southern Oscillation (ENSO) modulating the strength of this teleconnection which largely contributes to the subseasonal predictability of the North Atlantic. More generally, the tropical background state and variability are essential to induce subseasonal predictability of the northern hemisphere circulation, especially in winter, provided that the climate phenomena supporting the teleconnection, such as the atmospheric upper-level jet are adequately simulated (Yamagami and Matsueda, 2020). The stratosphere is another key precursor to the variability and predictability of the wintertime northern hemisphere circulation (Domeisen et al., 2020). A correct initialization (Zuo et al., 2016), together with a good representation of the stratosphere-troposphere coupling (Kolstad et al., 2020) accordingly contributes to skillfully forecast the NAO.

Other studies have focused on the predictability conditioned by the wintertime weather regimes occurring at initialization time. Based on a specific set of weather regimes affecting North America, Vigaud et al. (2018) demonstrated the capacity of the ECMWF subseasonal forecast system to successfully predict some of them up to two weeks. Robertson et al. (2020) built on this study to emphasize the value of this weather regime approach for identifying forecasts of opportunity over North America, with high skill up to 30 days ahead for specific events or seasons. The flow-dependent variations of the subseasonal forecast skill over Europe was also evidenced (Ferranti et al., 2018), with a relatively good capacity of the ECMWF system to reproduce the preferred transitions between weather regimes. Ferranti et al. (2015) identified differences in medium-range weather forecast performances conditional to the regime flow in the initial conditions with initial NAO- states leading to more skillful forecasts. Beyond approaches based on weather regime prediction, Minami and Takaya (2020) recently found that Northern Hemisphere 500 hPa geopotential height was more predictable when following strong negative initial AO, due to an eddy-zonal flow feedback that contributes to persist this mode of atmospheric variability. This study emphasizes the role played by large-scale extratropical atmospheric dynamics in subseasonal predictability, on top of tropical and stratospheric precursors.

Our main goal here is to further explore the relationship between the circulation flow present in the forecast initial conditions, hereafter initial weather regimes, and subseasonal predictability of the 2m-temperature in winter over a broad North Atlantic European domain. In this study we analyze jointly the ECMWF forecast system, and the most recent CNRM (Météo-France)



**Table 1.** Characteristics of the CNRM and ECMWF subseasonal reforecasts

| Characteristic | CNRM | ECMWF |
|---|---|---|
| **Model** | CNRM-CM6-1 HR (Voldoire et al., 2019) | ECMWF IFS CY43R3 |
| **Horizontal resolution** | TL359 (~50 km) | Tco639 (~15 km) up to day 15, Tco319 (~31km) after day 15 |
| **Vertical resolution** | 91 levels up to 0.01 hPa | 91 levels up to 0.01 hPa |
| **Ocean resolution** | 0.25°, 75 levels | 0.25°, 75 levels |
| **Reforecast ensemble size** | 10 | 11 |
| **Reforecast frequency** | Thursdays | Bi-weekly |
| **Reforecast system** | fix | on-the-fly |
| **Atmospheric/Land initial conditions** | ERA5 (Hersbach et al., 2020) | ERA-Interim (Dee et al., 2011) |
| **Ocean/sea-ice initial conditions** | Mercator Ocean International | ORAS5 |
| **Ensemble generation** | Stochastic dynamics (Batté and Déqué, 2016) | Perturbed initial conditions (Singular vectors, Ensemble Data Assimilation) + Stochastic physics (SPPT and SKEB schemes) |

subseasonal forecast system, launched in October 2020. The next section presents these forecast systems, as well as reference data and methods adopted in this study. The main results are then developed in a dedicated section. The last section provides
concluding remarks and prospects.

## 2  Data and methods

### 2.1  Forecast systems

Subseasonal forecasts delivered by CNRM have been routinely feeding the S2S database (Vitart et al., 2017) since 2015 with forecasts issued every Thursday. Lately, the CNRM forecast system version 1 (Ardilouze et al., 2017) has been superseded
by a version 2 used in this study. Unlike the ECMWF extended range forecast system (which also feeds the S2S database), the CNRM upgraded system has been designed for research purposes and is not intended for operational aspects. Since the ECMWF system is often acknowledged as the most skillful system in several intercomparison comparison studies (e.g. Zheng et al., 2019; Specq et al., 2020), it will serve as a benchmark in the present work to assess the performance of new CNRM system. The main characteristics of both forecast systems are described in table 1.



In this manuscript, 'reforecast' and 'forecast' indistinctly refer to retrospective forecasts, also named 'hindcast' in other studies. The comparison of ECMWF and CNRM prediction systems is facilitated by their comparable reforecast ensemble size and a common 20-year reforecast period. Here, we consider the December-to-March extended winters from 1997/1998 to 2016/2017.

However, because of different reforecast designs, initial dates do not exactly match between the two systems. This issue is
addressed as follows. We first select for each winter 16 consecutive CNRM start dates (i.e. Thursdays) after November 13th, so that week 3 and 4 are always included within the December to March 4-month period. Then for each of these 320 (16x20) CNRM initial dates, we pick the closest date among the available ECMWF initial dates. Since ECMWF forecasts are issued twice a week, the resulting date from this selection either matches the CNRM counterpart or precedes/follows it by no more than two days, depending on the reforecast year. Note that each reforecast is evaluated against the corresponding reanalysis
dates, to ensure a perfectly fair inter-model comparison.

Forecast and observed daily anomalies are considered rather than full fields, in order to remove the model bias : for the $n^{th}$ forecast ($n \leq 16$) of a given winter, daily anomalies are computed by substracting the daily climatology corresponding to the mean of the $n^{th}$ forecasts of the 19 other winters.

In this study, we follow a frequently used convention in the S2S community to define weekly lead times (e.g. Vitart, 2004;
Specq et al., 2020; de Andrade et al., 2021) : week 1 goes from day 5 to day 11, week 2 from day 12 to 18, week 3 from day 19 to 25 and week 4 from day 26 to 32.

For the composite analysis described in section 3.3.1, in addition to the forecast systems, we make use of a 300-year long pre-industrial simulation (known as piControl) of the same model used in the CNRM system, namely CNRM-CM6-1-HR, and performed in the framework of the Coupled Model Intercomparison Project Phase 6 (CMIP6, Eyring et al., 2016). This
experiment is useful to assess the behavior of the model internal variability without any drift from initial conditions nor forcing interference stemming either from initialization or volcanic and anthropogenic aerosols as well as greenhouse gases emissions. Additionally, this simulation provides enough years to work on densely populated composite samples, thereby ensuring an enhanced robustness of the results.

## 2.2 Reference dataset and forecast skill metrics

The ERA5 reanalysis (Hersbach et al., 2020) serves as the reference for daily sea-level pressure and daily-mean 2-meter temperature. This reanalysis, although resulting from a model output, assimilates a wide array of observations, and will therefore be considered as our observational reference. For simplicity, we will use the term "observation" - albeit abusively - to refer to ERA5 in the rest of the manuscript. Because ERA5 and ECMWF reforecasts are derived from two versions of the same model, one may object that ERA5 is not a suitable reference for this study. We have thus compared a few results obtained with
the JRA-55 reanalysis (Kobayashi et al., 2015) as reference instead of ERA5. Given the very limited differences found (not shown), we have chosen to pursue with ERA5 only.



A common score to evaluate a subseasonal forecast system is the point-wise Pearson correlation between the ensemble mean forecasts and the corresponding observations over the entire reforecast period. Grid-point time correlation is a classic deterministic score, whose significance is here determined by a two-sided Student t-test at the 95% confidence level.

In order to evaluate the skill of an individual forecast, we also compute the anomaly correlation coefficient (ACC) which shows the level of spatial agreement between the forecast and observed patterns. This is performed over a domain covering Europe (hereafter EUR, 12°W,41°E,34°N,65°N). The domain boundaries are displayed on the map of figure 1. For ACCs, the significance is obtained by a bootstrapping method applied to the ensemble members of the forecasts: we compute the ACCs of 100 draws among the 10 (11) members of the CNRM (ECMWF) forecast and consider the forecast skillful if at least 95% of
the 100 ACCs exceed zero. The root mean square error (RMSE) measuring the distance between the ensemble mean forecast and observation regardless of the sign of the anomaly has also been computed for individual forecasts, and normalized by the interquartile range of the observation. However the RMSE normalization method is arbitrary and this score has only been used to confirm a result found with the ACC in section 3.1.

In addition to deterministic scores, the ensemble forecasts can be evaluated by means of probabilistic skill metrics. The
continuous ranked probability score (CRPS) is the quadratic difference between the cumulative distribution function (CDF) of an ensemble forecast and the empirical CDF of the observation. The smaller the CRPS, the more accurate the forecast. Let $F(x)$ be the forecast CDF for the variable $x$ (e.g. weekly-mean 2-meter temperature), and $y$ the corresponding observation, then the analytical expression of the CRPS is :

$$CRPS = \int_{\mathbb{R}} (F(x) - \mathbb{1}(x \geq y))^2 dx \tag{1}$$

where $\mathbb{1}$ is the indicator function.

It is also insightful to compute a continuous ranked probability skill score (CRPSS) for a dynamical forecast system by comparing its CRPS ($CRPS_f$) with that of a climatological forecast ($CRPS_c$) so that :

$$CRPSS = 1 - \frac{CRPS_f}{CRPS_c} \tag{2}$$

CRPSS ranges between $-\infty$ and 1, 1 corresponding to a perfect forecast. Negative CRPSS values indicate that dynamical
forecasts are less accurate than climatological forecasts. In this study, we consider 16 forecasts per winter over 20 years. Therefore, for the $n^{th}$ ($n \leq 16$) forecast of a given year, the corresponding climatological forecast consists in a 19-member ensemble forecast grouping the $n^{th}$ forecasts of the 19 other years. To take into account the differences in ensemble size between the forecasts and their corresponding climatological forecasts, a so-called 'fair' version of the CRPSS is computed, via an unbiased estimator for the score that would be obtained as the ensemble size increases to infinity (Ferro et al., 2008;
Ferro, 2014).





## 2.3 Weather regimes and NAO index

The computation of weather regimes is performed on the ERA5 1979-2017 extended winter, i.e. the months of November to March (hereafter NDJFM). It consists in a k-means clustering of daily maps of sea-level pressure (SLP) anomalies of the North-Atlantic Europe (NAE) domain defined by the boundaries 90°W,30°E,20°N and 80°N. In order to facilitate this

clustering, an Empirical Orthogonal Function (EOF) analysis is applied to the SLP anomaly maps, for which the 19 leading modes are retained, explaining more than 90% of the SLP variance. The four resulting clusters correspond to the typical North-Atlantic weather regimes widely described and used in the literature (e.g. Michelangeli et al., 1995). By decreasing order of frequency, these regimes are identified as positive phase of the North-Atlantic oscillation (NAO+), Scandinavian blocking (BLO), negative phase of the North-Atlantic oscillation (NAO-) and Atlantic ridge (AR). Each winter day of the reanalysis

and the model simulations is then assigned to the weather regime for which the root mean square distance between the regime centroid and the map of SLP anomaly is minimal.

The assessment of teleconnections is facilitated by the use of a NAO index that quantifies this oscillation. Here, it is calculated as the normalized time series of the first principal component, resulting from the projection of the daily ERA5 SLP anomaly field on the leading EOF. For further robustness and because there are multiple ways to define the NAO (Pokorná and Huth,

2015), a comparison is made with another NAO index computed independently by the U.S. National Oceanic and Atmospheric Administration (NOAA) (NOAA Climate Prediction Center NAO index, 2020) on 500 hPa geopotential height fields from the NCEP/NCAR reanalysis and using a different method (Barnston and Livezey, 1987). Despite the many differences between the two daily NAO indices, their correlation for NDJFM 1979-2017 is as high as 0.77.

## 3 Results

In this section, we start with a general skill assessment to obtain a compared overview of the model ability to predict 2-meter temperature at the subseasonal horizon. The second and third subsections address the question of flow-dependence and the consequences on the forecast skill.

### 3.1 Skill of the subseasonal forecast systems

#### 3.1.1 Northern hemisphere assessment

The pointwise Pearson correlation between forecasts and observation is shown for week 1 to week 4 forecast times in figure 1.

It clearly shows for both systems the sharp decrease of skill after week 1, and also the better performance of the ECMWF system for the 4 weeks. This result was somehow expected given the much finer spatial resolution of the ECMWF system (Vitart, 2017). The skill difference could also originate from the better fit between the ECMWF forecast system and the ERA-Interim initial conditions, derived from another version of the same IFS model. Nonetheless, discussing the impact on skill of

ECMWF and CNRM modelling and forecasting strategies is out of the scope of this study.

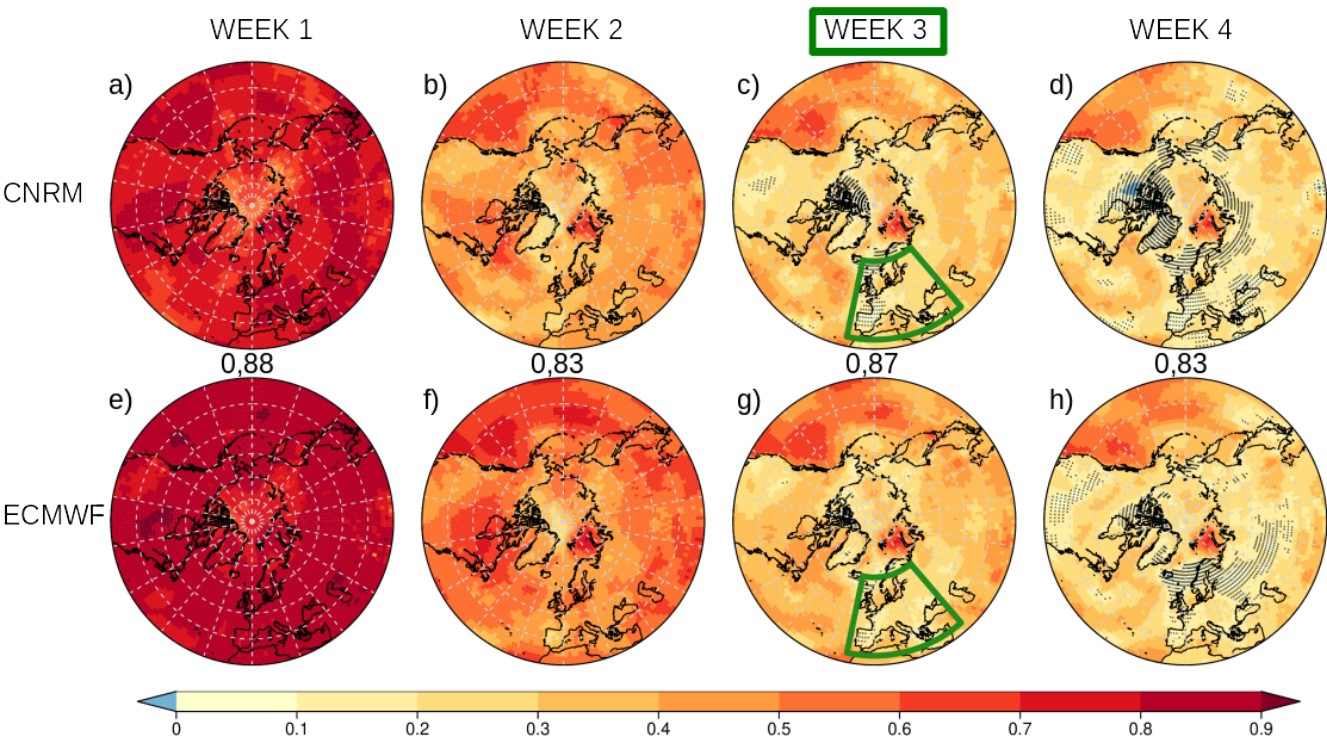

**Figure 1.** Correlation between week 1 to week 4 2-meter temperature forecasts and the corresponding observation for CNRM (a to d) and ECMWF (e to h) forecast systems. Stippling indicates grid points where correlation is **not** significantly positive at the 95 % confidence level. The numbers show the spatial correlation between CNRM and ECMWF maps for each week. Green boxes indicate the focus region (EUR) and forecast lead-time (week 3) targeted in section 3.1.2.

For both models, the correlation at week 3 remains positive for large parts of the Northern hemisphere extratropics, albeit weakly over continents. At week 3 and 4, the ECMWF forecasts still show significant correlation over most of Europe, while this is only true over Eastern Europe for CNRM. Overall, while ECMWF exhibits higher skill than CNRM, the large-scale patterns of gridpoint correlation are strikingly similar between both models, as confirmed by the high values of spatial correlations reported on Figure 1.

However, positive correlations do not guarantee that these forecasts are more useful than a naive forecast. To document this issue, we compare the CRPS probabilistic score with that of a climatological forecast, by means of the fair CRPSS (see section 2.2). On these maps (fig. 2), white and blue shadings indicate regions where the forecasts do not perform better than the climatology. This score highlights the much better performance of ECMWF over CNRM as early as week 2. The skill patterns look like those found in the correlation analysis, but they are more drastic. For example at week 3, over Europe, the CNRM system shows only remnant skill near the Baltic sea, and the ECMWF over the North of the continent as well as a limited





portion of Central Europe. The contrast between the two systems is even more striking over North America. The comparatively poor CRPSS of CNRM could be the consequence of a lack of ensemble spread, resulting in a too narrow distribution of forecasts, which denotes overconfident predictions. The complementary analysis shown in Appendix A, which compares the
intra-ensemble standard deviation of the two systems from week 1 to week 4, tend to confirm this hypothesis.

Interestingly, the systems remain relatively skillful over the Mediterranean sea but also the sea of Okhotsk, the Kara, Barents and Labrador seas, and, to a certain extent, the Baltic sea. This could be a consequence of persisting sea-surface temperature (for the Mediterranean) and sea-ice extent (for the Arctic and North Atlantic seas) anomalies leading to enhanced subseasonal predictability to the near-surface atmosphere (Chevallier et al., 2019; Bach et al., 2019), although indisputable evidence would
require a dedicated study. From now on, our work focuses on the predictability of week 3 only.

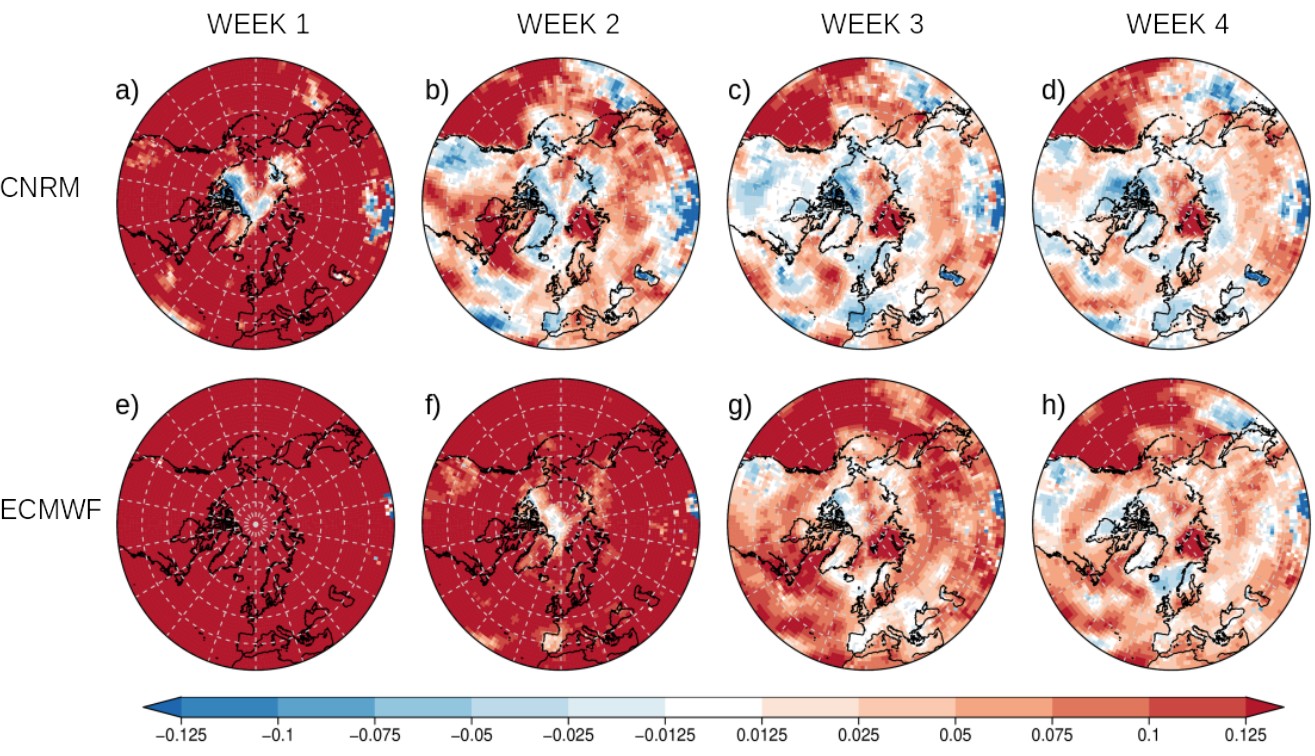

**Figure 2.** Fair CRPSS for week 1 to week 4 forecasts for CNRM (a to d) and ECMWF (e to h) forecast systems, against climatological forecasts (see text). Red shades indicate that the actual forecasts are more skillful than the climatological counterparts.

### 3.1.2  Focus on Europe

The forecast skill of EUR 2-meter temperature is assessed from the 320 reforecasts at week 3 for both systems, by means of the ACC. This score varies considerably between dates. Thus, in order to investigate the degree of consistency between models





forecast skill, Figure 3 plots the distribution of ECMWF ACC against CNRM ACC over EUR, for each of the reforecast
dates. Dots depict the 320 reforecasts and filled contours the corresponding probability density function. Red dots show the
reforecasts where ACCs are significant at the 95 % level for both systems. This distribution is fairly symmetric, albeit slightly
skewed towards higher values for ECMWF, which is consistent with results found in the previous section. This is also revealed
by the mean and median points (black and grey triangles, respectively), located slightly above the $y = x$ identity line. The
standard deviation of ACCs is similar (0.42 for CNRM vs. 0.40 for ECMWF). More interestingly, the correlation between
CNRM and ECMWF ACCs reaches 0.52. The correlation is even higher (0.61) when considering the RMSE of the individual
forecasts instead of the ACC (not shown). The scatter plot also reveals that the most skillful concurrent forecasts (red dots)
are less scattered and more grouped along the $y = x$ identity line than other forecasts more spread out. They correspond to
the maximum of the probability density function, figured in green and yellow shades This suggests that high skill forecasts
contribute more to the correlation than low skill counterparts counterparts. In other words, CNRM and ECMWF systems are
more prone to issue concurrently good forecasts than concurrently poor ones.

The synchronicity found in the level of skill between the CNRM and ECMWF week 3 forecasts therefore indicates the
existence of a common source of predictability concerning the EUR region.

We now investigate the distribution of skillful forecasts along the 20 year period considered in this study. The barplots in
figure 4 show a relatively consistent trend and interannual variability : the number of yearly skillful forecasts for ECMWF, in
red, is significantly correlated to that of CNRM, in blue (r=0.61). Linear trends have a similar slope, as reported on the figure.
We reprocessed figures 3 and 4 after removing a linear trend derived from the DJFM ERA5 2-meter temperature averaged over
the Europe domain. We found no significant changes in the ACC distribution and correlation (0.519 instead of 0.521), nor in
the interannual variability of skillful forecasts (not shown). Limited changes in the number of significantly positive ACCs per
year and per forecast system before and after detrending confirm the minimal influence of the warming trend on the forecast
skill.

In any case, 2009-2010 stands out as the winter with the maximum number of skillful forecast of the 20-year period for
both CNRM and ECMWF systems either considered jointly (green bars) or separately (blue and red bars). Since that winter
is characterized by a record-breaking negative NAO index (Cattiaux et al., 2010), we have computed the correlation between
the yearly number of skillful forecasts and a winter mean NAO index (December-to-March) derived from our daily NAO index
datasets. The correlation is not significant if the NAO index is computed by averaging daily NAO indices (not shown). However,
when computed as the mean of daily absolute values of the ERA5 NAO index (brown broken line in fig. 4), the correlation
found is significant. This is also true with the NOAA NAO index, with $r$ ranging from 0.44 to 0.66. This result suggests that
S2S EUR forecasts are more frequently skillful during winters characterized by a strong NAO index, either positive or negative.

Therefore, next section focuses more specifically on the relationship between forecast skill and weather regimes.



**Figure 3.** Scatter plot (dots) and probability density function (contours) of ECMWF ACC in function of the corresponding CNRM ACC for each of the 320 wintertime reforecasts of EUR 2-meter temperature at week 3. Red dots mark ACCs significant at the 95 % confidence level for both CNRM and ECMWF. The black and grey triangles correspond to the mean and the median point respectively, and the black solid line to the $y = x$ identity line

## 3.2 Relationship between forecast skill and initial weather regime

We now consider the first 4 days after initialization as a relevant time window to discuss about initial weather regime. We argue that the choice of using 4 days instead of the single first day allows more robustness, since the latter may sometimes lie



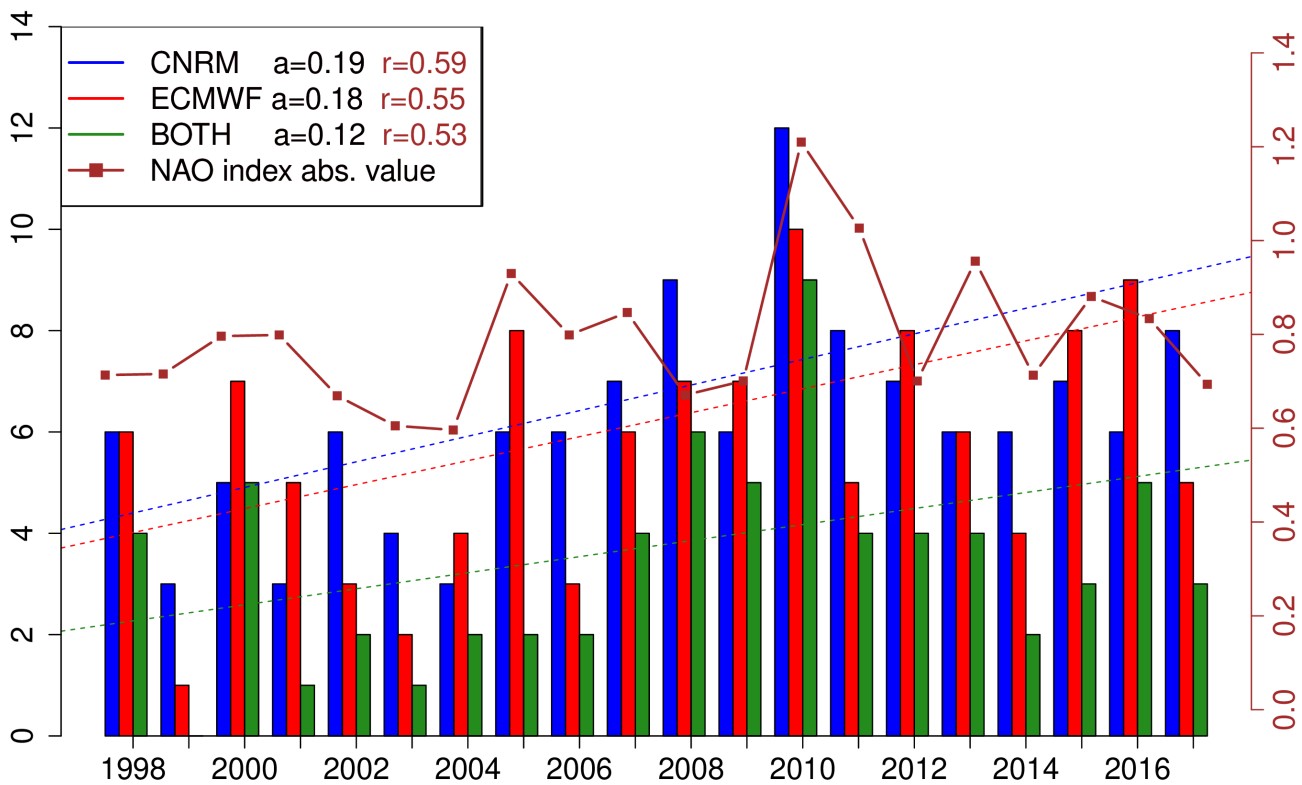

**Figure 4.** Yearly number of skillful forecasts for CNRM (blue), ECMWF (red) and both systems (green) computed on EUR week 3 temperature forecasts. The dashed lines mark the respective linear trend whose slope value 'a' is reported in the legend. The brown broken line shows the absolute value of the winter NAO index derived from ERA5 (see text). The 'r' values reported in the legend correspond to the correlation of this index with the yearly number of skillful forecasts.

in between two different regimes. This 4-day window is also consistent with the S2S convention that defines the first forecast week as starting from day 5 onwards (see section 2.1).

We count the occurrence of each weather regime assigned to the first 4 days of the forecast members, among the 68 EUR forecasts out of 320, that are concurrently skillful for CNRM and ECMWF.

In this sample of 68 skillful reforecasts, the frequency of initial NAO+ days is significantly higher, and that of initial BLO days lower than in the 252 other reforecasts, for both forecast systems (Table 2). The frequency of NAO- initial days is also higher in CNRM but not significantly for ECMWF.

**Table 2.** Initial weather regime frequency in % of skilful forecasts over EUR. Numbers in parentheses indicate the frequency for all the other forecasts and bold characters highlight where these frequency are significantly different at the 95 % confidence level as determined by bootstrap.

| Weather regime | NAO+ | BLO | NAO- | AR |
|---|---|---|---|---|
| CNRM | **39.2 (26.5)** | **20.4 (30.5)** | **29.6 (22.6)** | 16.3 (20.4) |
| ECMWF | **38.2 (27.8)** | **17.5 (26.8)** | 27.8 (23.1) | 16.5 (22.3) |

If skillful forecasts tend to start more frequently with NAO conditions, we would like to verify the reciprocal, i.e. how skillful the forecasts starting with NAO conditions are. To this end, instead of subsampling the forecasts according to their level of skill, we now cluster the 320 forecasts in 4 groups determined by their initial weather regime and compare the mean skill evolution along the forecast time for each of these clusters (fig. 5). We define the initial weather regime of each forecast as the regime with the greatest number of occurrence during the 4 initial days.

Here the significance level is obtained by means of a bootstrapping method. More precisely, for a cluster of size $N$, a probability density function of mean ACC is built out of 1000 draws with replacement of $N$ forecasts within this cluster. The forecast is then considered significant if the first percentile of the distribution is positive.

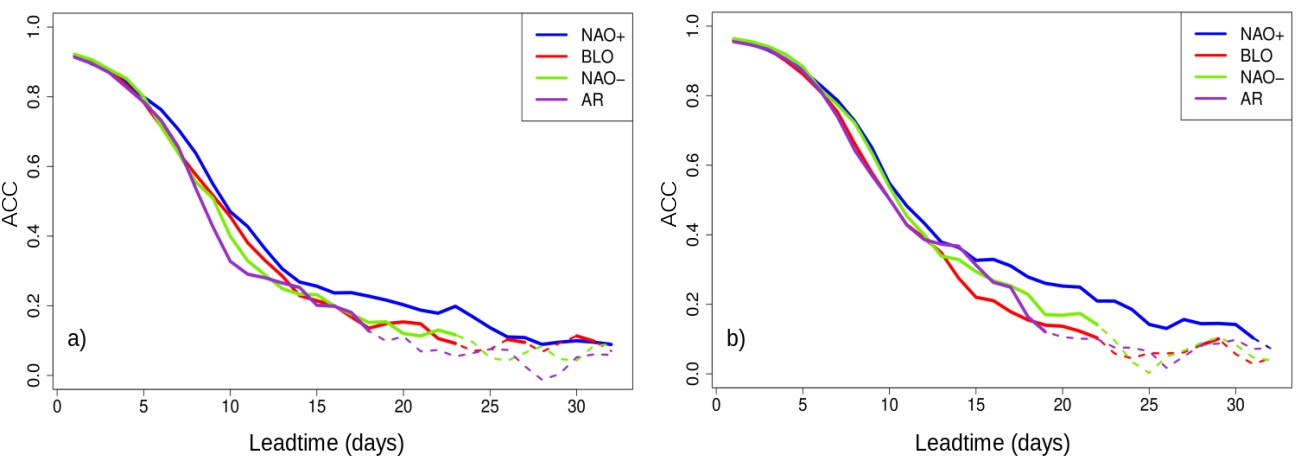

**Figure 5.** Mean ACC evolution with forecast time over Europe by initial weather regime for (a) CNRM and (b) ECMWF. Solid lines indicate values significantly positive at the 99 % confidence level.

For both systems, the mean ACC of the forecasts initialized in NAO+ conditions becomes higher than those initialized with other regimes from day 15 onwards. The difference vanishes past day 25 for CNRM but not for ECMWF. Finally, the ACC



remains significantly positive until the end of the forecast period in both systems although a positive ACC does not necessarily imply that the forecasts are useful, as discussed in section 3.1. It is also interesting to notice that NAO- conditions do not lead to particularly skillful forecasts at week 3 for CNRM, as could have been expected from table 2, and that models agree upon AR being the worst initial weather regime in terms of temperature subseasonal predictability over Europe, since the mean ACC of forecasts initialized thereby are no longer significant past day 18 or 19.

The next question arising from the previous results is the evolution in time of the regime frequency among these forecasts initiated under NAO+ or NAO- conditions. The stacked bar plots on figure 6 illustrate this evolution. Note that the residual non-NAO+ (resp. non-NAO-) regimes showing in the 4 initial days simply result from our clustering method based on the predominant regime counted within the 4 initial days of all ensemble members, thereby leaving some room for the occurrence of other weather regimes. We find that despite a rapid decrease of the NAO+ regime proportion with forecast time, it remains

slightly larger than the climatological one at week 3. A similar but more pronounced result is found for NAO-. This suggests that the NAO regimes are persistent in the forecasts although this cannot be ascertained at this stage since no statistical significance test has been performed here, and furthermore, all the ensemble members are pooled together, which conceals the transitions between weather regimes.

Before exploring further the causes of the above results, we need to address in more detail the question of regime persistence.
So far, every forecast or observed day has been assigned with one of the 4 weather regimes, regardless of the day-to-day variability of the regime sequence. Such variability occurs when the spatial distribution of high and low pressure systems of a given day does not match well any of the canonical weather regimes, or corresponds to a transition between two of them. To overcome this issue, we have defined a fifth category (called 'NONE') assigned to days outside any sequence of 3 or more days belonging to the same weather regime. Excluding the forecasts initialized with the predominant 'NONE' category results

in 4 smaller clusters of forecasts. Nonetheless the mean ACC evolution is not dramatically changed, and the ACC dependence on the initial conditions lead to the same hierarchy of weather regimes as can be seen in Appendix B. Similar conclusions can be drawn regarding the weekly evolution of regime frequency after taking the 'NONE' category into account. Given the limited impact of the screening based on regime persistence, the following sections rely on the original daily weather regime assignment, i.e. without the 'NONE' category.

**3.3   Evidence of a lagged teleconnection**

**3.3.1   Composite analysis**

The previous section has pinpointed a slight distort in the weather regimes distribution at week 3 for forecasts initialized in NAO conditions. For a broader comprehension, we thus compute the spatial composites of week 3 anomalies for this subset of forecasts for sea-level pressure (fig. 7a and b) and 2-meter temperature (fig. 7e and f). Given that 87 (93) forecasts out of 320

are concerned for CNRM (ECMWF) and that each of them comprises 10 (11) members, the composite maps result from the average of $n=870$ ($n=1027$) single realizations. The ECMWF and CNRM composites show some similarities over the Atlantic



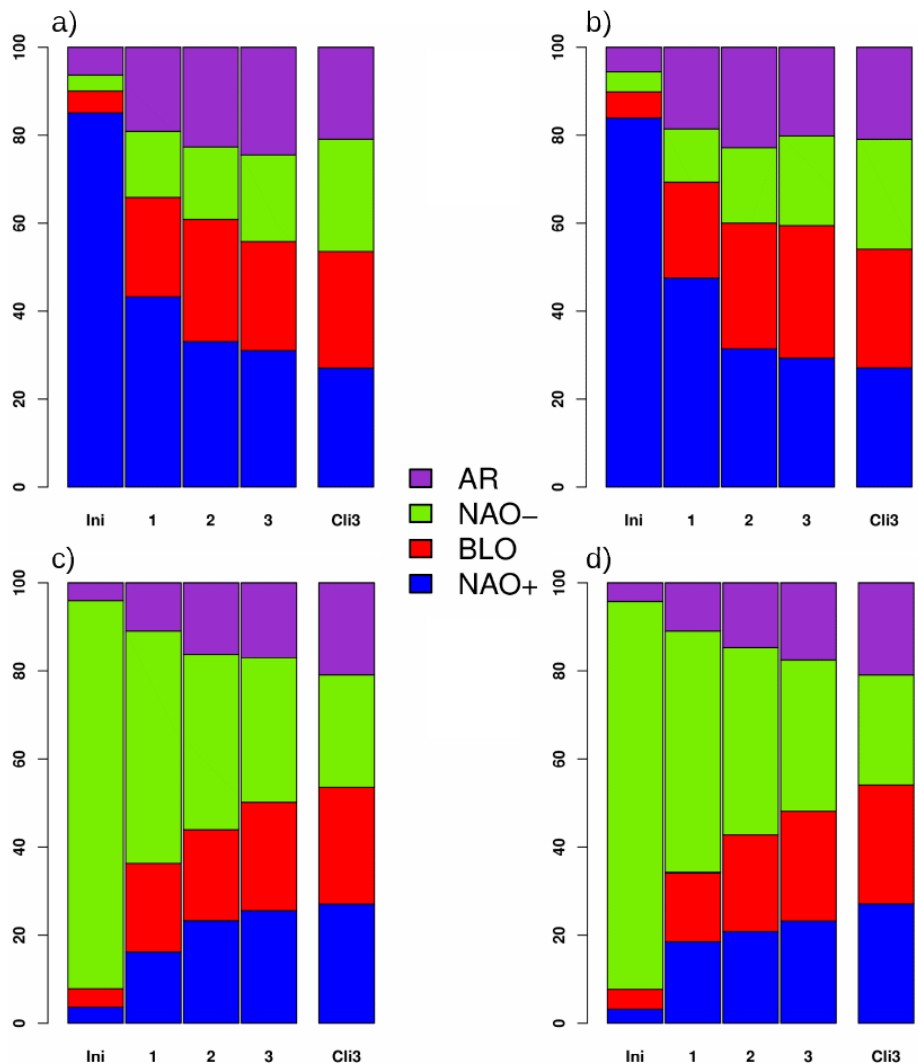

**Figure 6.** Weekly evolution of regime frequency among forecasts initialized in NAO+ (a) or NAO- (c) conditions for CNRM. (b) and (d) same as (a) and (c) for ECMWF. The leftmost bar corresponds to the 4 initial days. The rightmost bar corresponds to the climatological frequency for week 3.

sector with a distinct low pressure anomaly over the Arctic and high pressure anomaly centers near the Azores archipelago. The temperature patterns are even closer to each other with a large scale warm anomaly stretching from Central Europe to Eastern Siberia and a cold anomaly over Canada, more pronounced near the Labrador Sea. The main difference between ECMWF and CNRM concerns the sea-level pressure anomaly over Europe, which barely reflects onto the temperature anomaly. If we consider the positive pressure anomaly over North Pacific, it may remind of the Arctic Oscillation (AO) loading pattern (e.g.



fig.1 in Thompson and Wallace, 1998), although this anomaly is not significant for CNRM, and more importantly not consistent with observations (see below).

The patterns found could be specific to the forecast systems, i.e. GCMs constrained by imposed initial conditions and external forcing affected by a strong anthropic imprint. To verify this hypothesis, we derive a set of single-member pseudo-forecasts from the CNRM-CM6-1-HR 300-year-long piControl simulation. For each simulated year, we extract sixteen 32-day time series starting every seven days from Nov. 13$^{th}$ to February 26$^{th}$, so as to mimic successive S2S forecast start dates. Among the 4784 resulting pseudo-forecasts, those having all 4 initial days assigned as NAO+ are sampled to compute sea-level pressure and 2-meter temperature anomaly composites (fig. 7c and g). In this case, it concerns $n$=579 realizations. We proceed likewise for the 1950-2017 ERA5 reanalysis (fig. 7d and h), in order to compare the realism of this behaviour with respect to observation.

The piControl composite shows broadly consistent patterns over the mid-Atlantic notwithstanding differences in terms of relative amplitude and extent of pressure anomalies. For temperature, the warm anomaly over Southeastern US is somewhat stronger than in the forecast systems. The amplitude of the ERA5 composite patterns is generally larger, which is at least partly explained by the reduced size of the composite sample ($n$=148). This observational composite shows a larger extent of the Atlantic high pressure belt also covering Southern Europe and central Asia, and conversely no high pressure anomaly over North Pacific, which tends to confirm that the similitude with a hemispheric positive AO pattern evoked earlier is a model artefact. In terms of temperature, the main difference is the greater extent of the warm anomaly over North America and the cold pattern near Bering strait, with respect to the forecast and piControl composites.

A similar composite analysis has been carried out with (pseudo-)forecasts initiated in NAO- (fig. 8). Here the pressure and temperature composites show even more similarities between forecasts and observation, in particular over the North-Atlantic-Europe region. Similar to NAO+, surface pressure patterns show more differences than temperature, in particular over East Siberia, West Pacific and North America. In the piControl composite, again, the patterns found are less intense but very consistent for temperatures, less so for surface pressure. One explanation for this reduced consistency could be that in the piControl time series, boundary conditions such as the ocean, sea-ice and stratosphere also influencing the atmospheric flow, have no reason to be coherent with observation, unlike the forecast composites initialized with reanalyzed atmospheric boundary conditions.

To summarize, this composite study reveals some significant agreement between forecast systems, unforced GCM and reanalysis as to prevailing atmospheric flow and near surface temperature anomalies during the third week following NAO conditions. This agreement is much better for negative than positive NAO, for temperature than pressure patterns, and for the North-Atlantic (pressure), Labrador, Europe and Siberia (temperature) regions.

### 3.3.2 Observational NAO index

At this point, our study has only considered a weather regime assignment based on a root mean square distance criterion but this method may conceal a wide array of a atmospheric situations. Here we make use of the NAO index that quantifies the



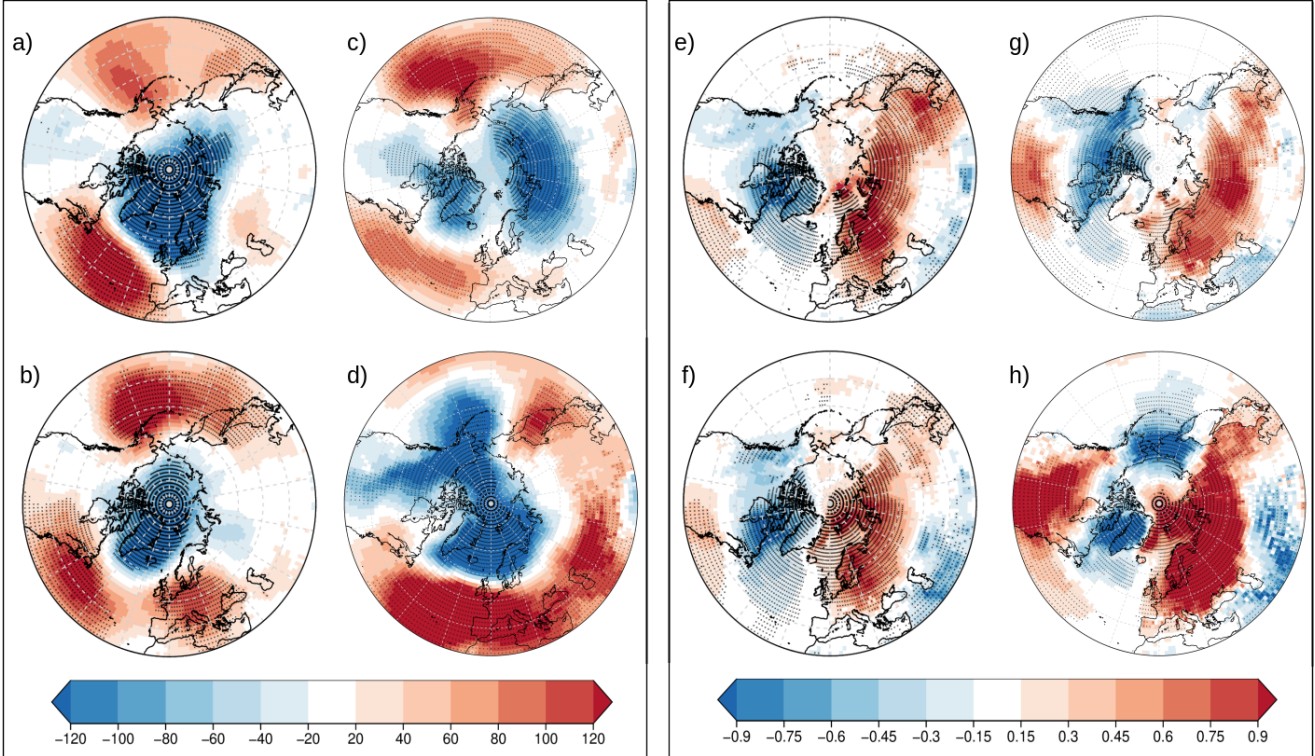

**Figure 7.** Composite anomaly of Week 3 sea-level pressure (in Pa) following NAO+ initial conditions in (a) CNRM forecasts (*n*=870) (b) ECMWF forecast (*n*=1027) (c) CNRM piControl pseudo-forecasts (*n*=579) and (d) ERA5 reanalysis (*n*=148). (e) to (h) like (a) to (d) for 2-meter temperature (in K) initial conditions. Anomalies statistically significant at the 95% level are stippled.

amplitude of the oscillation, and allows to identify periods of intense NAO+ or NAO- conditions. The composite analysis suggests that NAO initial conditions lead to NAO-like atmospheric flow. To verify this, we evaluate the extent to which the NAO index decorrelates with time in the observation. More precisely, the figure 9 depicts the correlation of the averaged day-1-to-day-4 NAO index with a sliding window of 7-day running mean NAO index. The grey line and envelope consider the 608 aforementioned time series (that is, 16 per winter of the 1979-2017 period) whereas the red counterparts only consider the 10 % characterized by the highest absolute value of the initial NAO index. Such screening selects the time series with an initial atmospheric flow characterized by intense NAO+ and NAO- conditions. Despite different datasets and methodology, the NAO decorrelation compares well to other studies when keeping the whole sample (Keeley et al., 2009) with a characteristic decorrelation time of 8 to 10 days. However, the decorrelation is much slower when considering only the subsample with intense NAO initial conditions. Keeley et al. (2009) also identified a similar "shoulder" or "rebound" in the NAO decorrelation function between 10 and 30 days and find it largely related to interannual variability, as opposed to intraseasonal. The overlap between confidence intervals indicates that the difference found is not significant beyond 10 days when the NAO index is

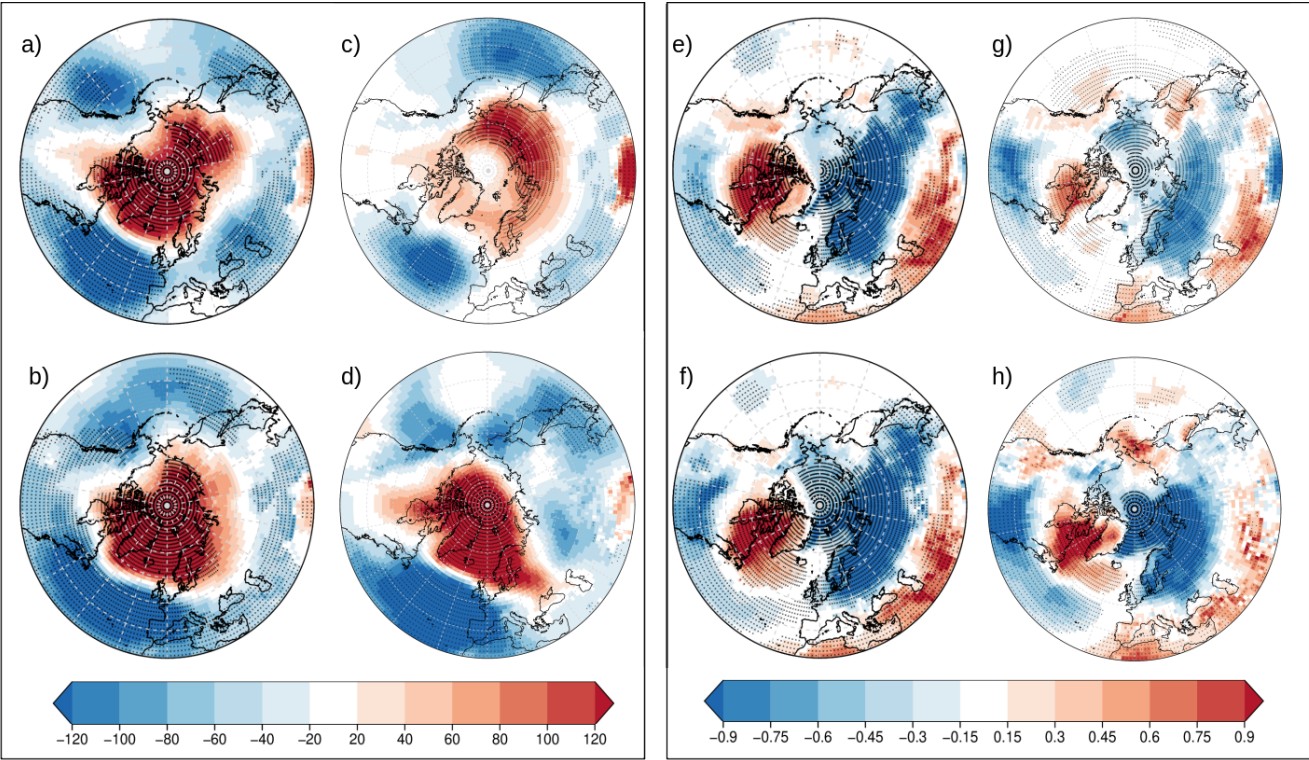

**Figure 8.** Same as fig. 7 for NAO- initial conditions.

derived from ERA5 sea-level pressure. However it remains largely significant for the NOAA NAO index based on 500 hPa geopotential height, in particular three weeks after initialization where the correlation peaks up. For that matter, the sensitivity of NAO persistence to the NAO index definition is consistent with previous findings (Domeisen et al., 2018). Regardless of

of the NAO index calculation method, our results provide observational evidence of a long-lasting persistence of NAO-like atmospheric flow in winter.

Finally, still with this observational subsample of ERA5 time series characterized by intense "NAO-like" initial conditions, we regress the week 3 pointwise 2-meter temperature onto the initial NAO index (fig. 10). Whether derived from ERA5 or NOAA, the patterns show similarities, with a stretch of positive correlation extending from South East US to Siberia with

maximum values near the Baltic Sea, and two negative correlation lobes over Greenland / Labrador sea, and from the tropical North-Atlantic to North Africa and the Middle-East.

Given that the spatial extent of these correlation patterns encompasses large parts of the northern hemisphere, we will now evaluate if NAO initial conditions of winter subseasonal forecasts could translate into enhanced prediction skill beyond Europe, and how this relates to the regression patterns described above.





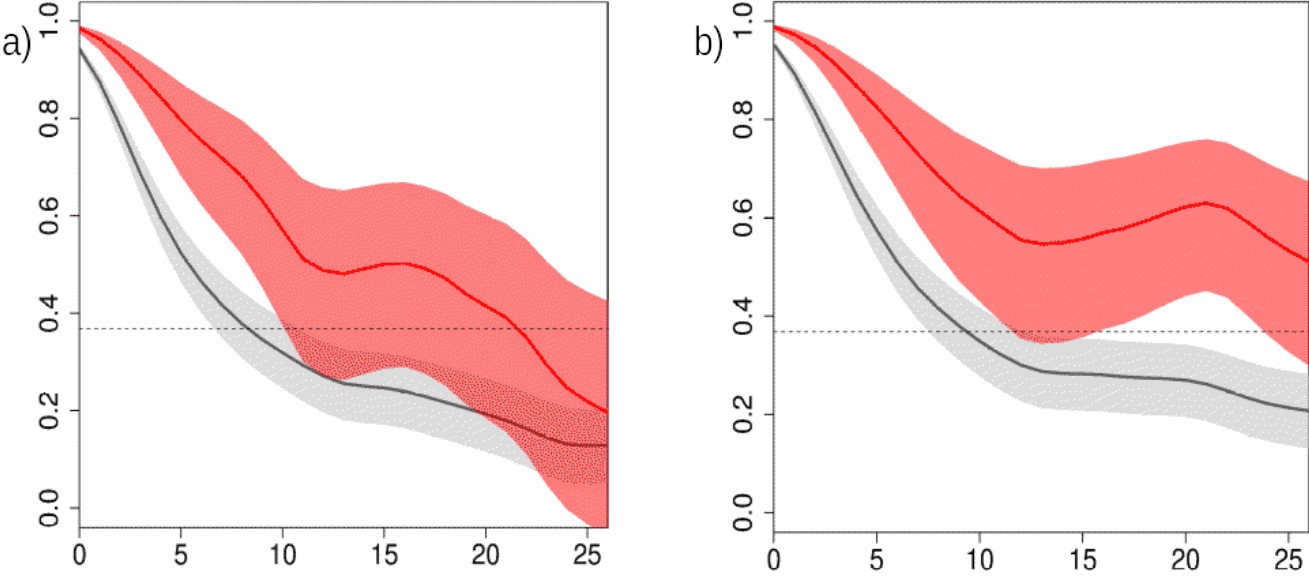

**Figure 9.** Correlation and 95 % confidence interval (solid line and envelope) of initial NAO index with weekly running mean NAO index derived from (a) ERA5 and (b) NOAA. Grey (red) shades consider the full sample (subsample with intense NAO initial conditions) of time series within the 1979-2017 wintertime period, as described in the text. The 1/e decorrelation threshold is marked with the dashed horizontal line.

## 3.4 Consequences on forecast skill outside Europe

In the previous section, we have identified a lagged NAO teleconnection prone to impact temperatures over a large fraction of the northern hemisphere. We now return to the forecast skill evaluation, but this time, we proceed to a subsampling of the reforecasts based on two conditions: the initial weather regime and its intensity. More precisely, we select all the reforecasts initiated in NAO+ and NAO- and evaluate their initial NAO index from the NOAA dataset. We then retain only the "initial NAO+" ("initial NAO-") reforecasts for which the initial NAO index belongs to the upper (lower) quartile of the distribution. Figure 11 shows the week 3 2-meter temperature correlation after subsampling and the correlation difference with respect to the full sample of reforecasts (see fig.1c and g) . The correlation patterns are patchier than in figure 1, because the sample size is considerably reduced, that is, 40 reforecasts instead of 320 for each system. Nonetheless, the correlation difference highlights a significantly increased skill over North-West Europe, and Central Siberia, as well as the Labrador seas and the South-East Mediterranean and Middle-East to a lesser extent. These regions match remarkably well with those concerned by the NAO lagged teleconnection highlighted in observations (fig. 10) and they are relatively consistent between CNRM and ECMWF systems. However, no improvement of skill is detected over South East US and off the US Atlantic coast, as could have been expected from the teleconnection patterns.

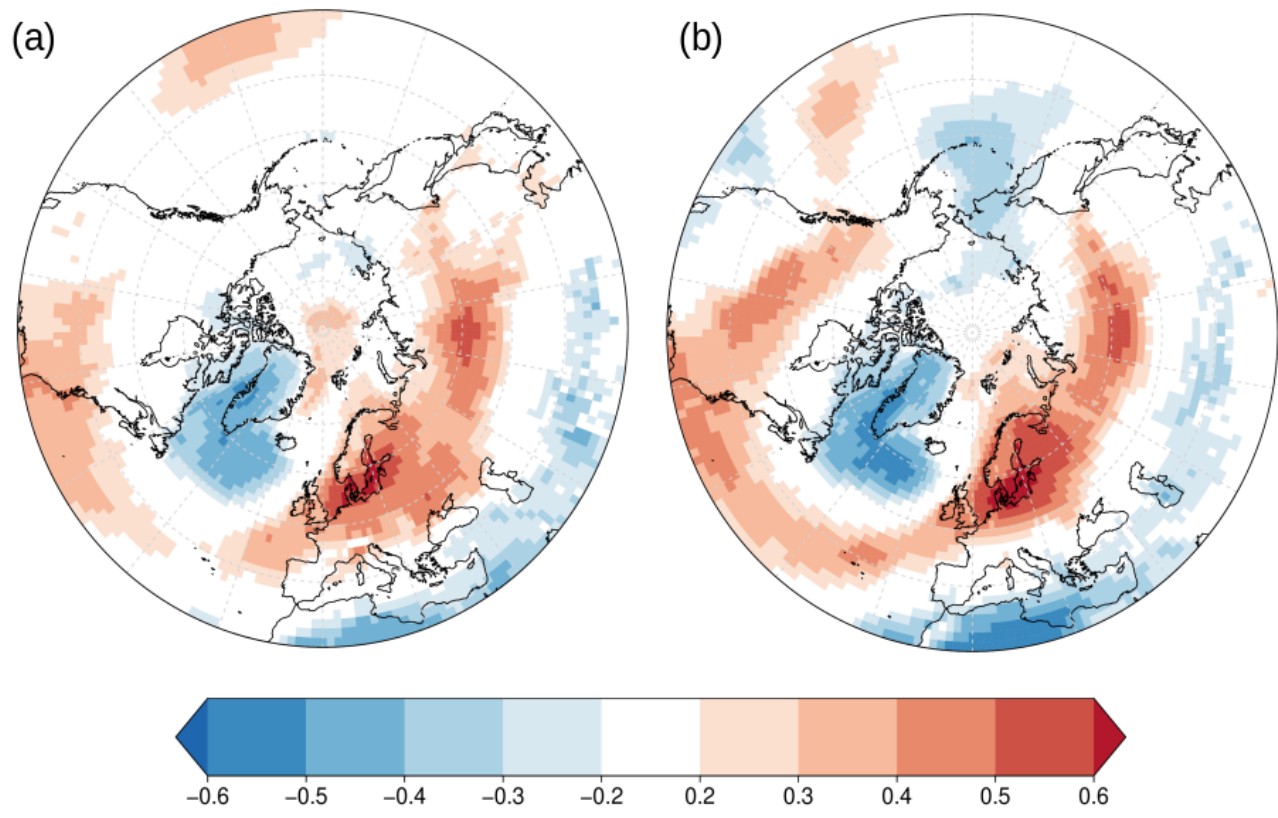

**Figure 10.** Correlation of ERA5 week 3 2-meter temperature with initial NAO index derived from (a) ERA5 or (b) NOAA. Only values significantly different from zero at the 99% confidence level are displayed.

Even if there is no one-to-one relationship between local increase or decrease of the prediction skill and the aforementioned

regression patterns, our study reveals consistent evidence that the forecast systems are capable of capturing the lagged NAO teleconnection to a certain extent. This provides additional sub-seasonal predictability at the continent scale, conditioned by the initial atmospheric flow.

## 4   Conclusions

The main objective of this study is to determine if the atmospheric circulation pattern in place at the time of initialization can

impact the subseasonal predictive skill of forecasts delivered by state-of-the-art forecast systems. This study focuses on winter



**Figure 11.** Correlation between week 3 2-meter temperature and the corresponding observation for the NAO initialized CNRM (a) and ECMWF (b) reforecasts. (c) and (d) depict the correlation difference (a) minus Figure 1(c) for CNRM and (b) minus Figure 1(g) for ECMWF. Stippling indicates significant values at the 95% confidence level.

northern hemisphere extratropics near-surface temperature reforecasts issued by the new CNRM subseasonal forecast system as well as the ECMWF extended-range forecast system.

A first general skill assessment shows that the CNRM system proves less skillful than the ECMWF counterpart when considering the first 4 weeks after initialization but performs reasonably well anyway. The ensemble spread of the CNRM forecasts is too weak over much of the Northern Hemisphere across all the prediction horizons, which likely penalizes this system in terms of probabilistic skill.

When considering the performances of individual successive forecasts over Europe, the level of skill at week 3 tends to vary concurrently for both systems, thereby suggesting that they benefit from a common and intermittent source of subseasonal



predictability. Since the European climate is known to be influenced by the North Atlantic Oscillation (NAO), a weather regime
approach has provided evidence that forecasts initialized during positive NAO phases are slightly more skillful over Europe
than those issued during the other 3 North-Atlantic weather regimes.

A composite analysis has shown that temperature and sea-level pressure anomalies typical of the positive (negative) NAO
regime tend to characterize the third week following the occurrence of such regime. This feature is well captured and compara-
ble to a certain extent in forecasts, pre-industrial climate simulations and observations, particularly for temperature anomalies.
The robustness of this time-lagged weather regime impact is further confirmed by the strong and persisting autocorrelation of
the upper and lower tail of the NAO index distribution.

Ultimately, we show that the subseasonal predictive skill over Europe is more pre-conditionned by intense NAO events,
either positive or negative, than by the prevailing regime at initialization. We also find that this flow-dependent skill concerns
mostly Northern Europe, but also central Siberia and regions surrounding the Labrador sea.

In a next study, it would be worth studying the atmospheric mechanisms involved in this NAO lagged teleconnection, and
the extent to which they are properly captured by forecast systems. Such an approach could bring insight about the reasons why
the NAO initiated forecasts do not show improved skill over most of Eastern North America, as could have been expected (Luo
et al., 2020). At least for the coastal area, recent findings from Roberts et al. (2021) indicate that the skill could be improved by
reducing the North-Atlantic sea surface temperature biases resulting from inadequate representation of mesoscale ocean eddies
in coupled models. Factors influencing the persistence of NAO+ and NAO- phases should also be investigated to go a step
further into the concept of flow-dependent "windows of opportunity" for subseasonal prediction. In particular the influence of
sudden stratospheric warming events on the occurrence and persistence of the NAO- regime has been evidenced (Domeisen,
2019). Hence, subseasonal forecasts issued after the onset of such events and characterized by a strong initial NAO phase could
be even more trustworthy, although this hypothesis would require a large reforecast dataset to be verified.

Another prospect for future works would be to evaluate the sensitivity of the results to the methodology. First, our strategy to
identify wintertime weather regimes, although widely referenced in literature, may not be optimal (Falkena et al., 2020). The
robustness of our results would be worth assessing when considering a different set of weather regimes. Then, the reforecasts
clustering strategy could be questioned. In particular, a distance threshold between sea-level pressure patterns in reforecasts
and the weather regime centroids could be applied in order to subsample only those reforecasts initiated in conditions very
close the canonical modes of atmospheric variability.

Finally, including more forecast systems for a multi-model approach would bring considerable interest but also a great deal
of additional complexity, given the many differences in the design of the S2S forecast systems.



## Appendix A: Comparison of the CNRM and ECMWF ensemble spread

Figure A1 shows the weekly evolution with leadtime of the intra-ensemble standard deviation of the 2-meter temperature for
the CNRM and the ECMWF subseasonal reforecasts. Since the CNRM ensemble size holds 10 members vs. 11 members for
ECMWF, only 10 members of the latter have been used to guarantee a fair comparison of the two systems. The week-by-week
differences (bottom row maps) help visualize that the ECMWF ensemble is more dispersive (red shades) than the CNRM
counterpart over the vast majority of the Northern hemisphere whatever the prediction horizon. Only the North Pole and to
a certain extent South Asia at longer lead times show more spread for CNRM. This lack of spread for CNRM is particularly
pronounced over high latitude continents but considering the slow evolution of sea-surface temperature, the lack of spread over
oceans is also meaningful and should not be overlooked.



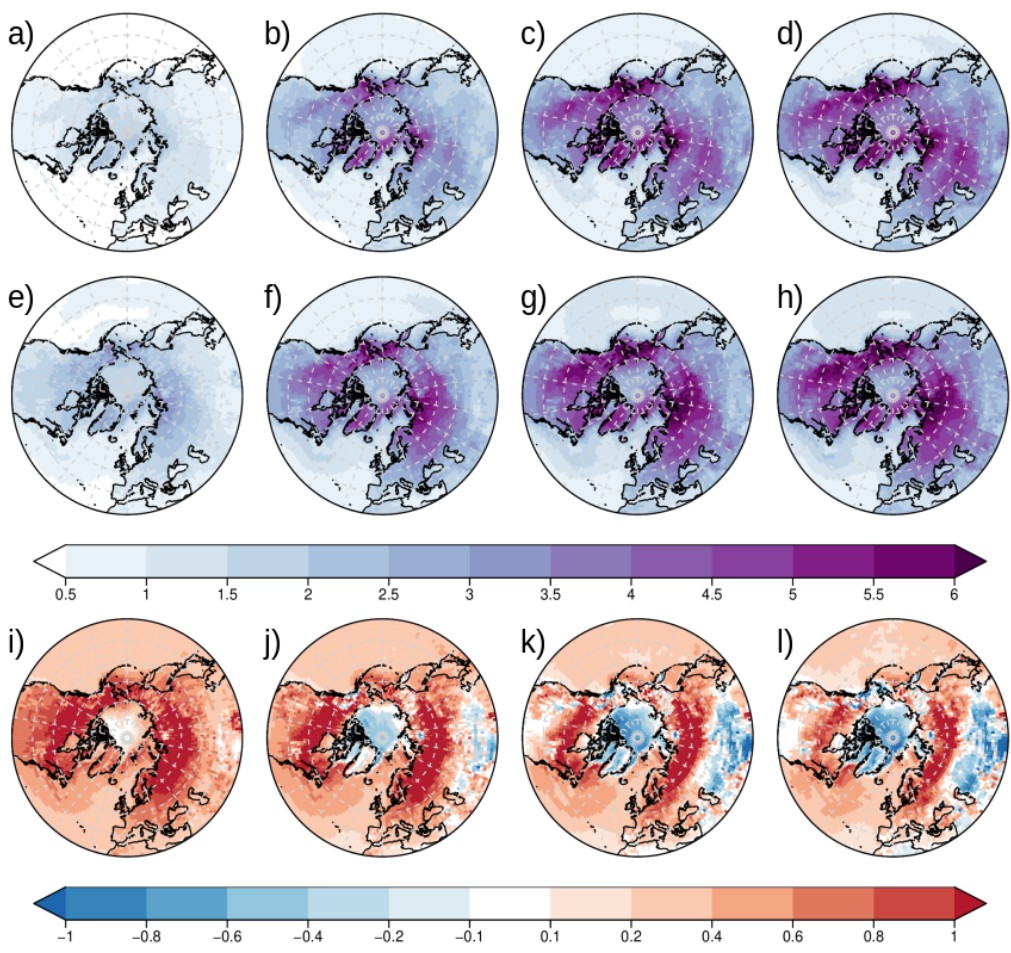

**Figure A1.** Ensemble standard deviation of 2-meter temperature for week1 to week for (a to d) CNRM and (e to h) ECMWF and week 1 to week 4 standard deviation differences 'ECMWF minus CNRM' (i to l). Differences not significant at the 95 % confidence level have been set to zero.





## Appendix B: Approach based on persisting regimes

We here provide the results obtained after taking into account the persistence of the weather regimes, resulting in a new category "None" (see section 3.2 for details). As can be seen in comparing figure B1 with 5 and figure B2 with 6, the results found are
very similar with or without this new category.

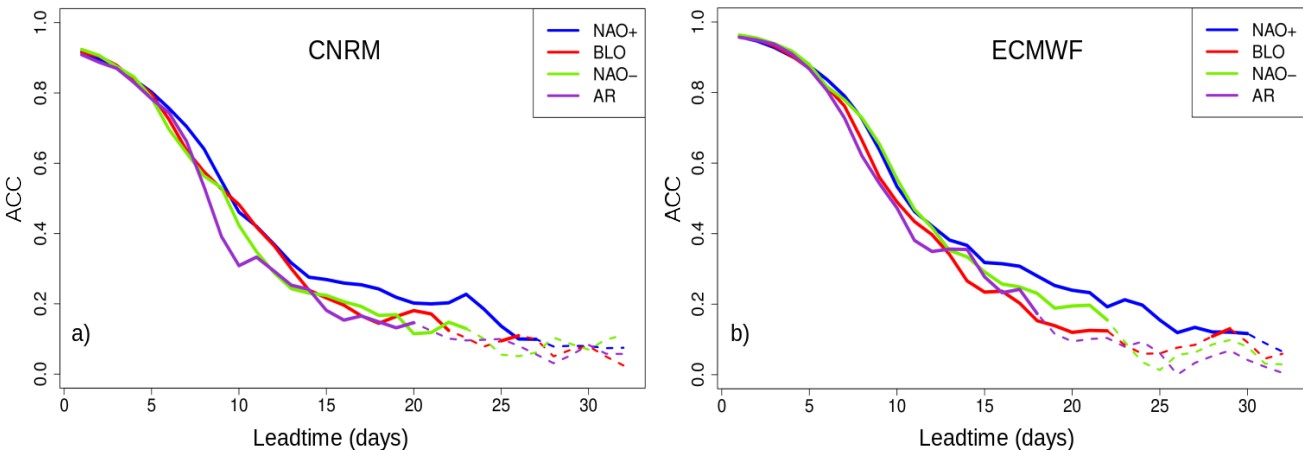

**Figure B1.** Like fig. 5 but with a fifth category 'None' including days outside any persistent sequence of a canonical weather regime







**Figure B2.** Like fig. 6 but with a fifth category 'None' including days outside any persistent sequence of a canonical weather regime



*Code and data availability.* The reforecast data used in this study is freely accessible from the S2S database (https://apps.ecmwf.int/datasets/data/s2s-realtime-instantaneous-accum-ecmf/levtype=sfc/type=cf/). The ERA5 reanalysis can be retrieved from the Climate Data Store (https://cds.climate.copernicus.eu/) and the CNRM-CM6-HR piControl simulation for CMIP6 from the Earth System Grid Federation platform (https://esgf-node.llnl.gov/search/cmip6/). The code for data analyses and plots is based on the free R software. Scripts are available
upon request.

*Author contributions.* Constantin Ardilouze has collected and analysed the data, contributed to the design and drafted this article. Damien Specq, Lauriane Batté and Christophe Cassou have equally contributed to the design and the critical revision of this article.

*Competing interests.* The authors declare that they have no conflict of interest.

*Acknowledgements.* This work is part of the 2020-2022 project ICHEPS, supported by the French National programme LEFE (Les Enveloppes Fluides et l'Environnement).



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
