# Peer review of "Flow dependence of wintertime subseasonal prediction skill over Europe"

_Weather and Climate Dynamics, 2021_

## Author Comment (AC1)

This study analyzes the impact of the initial state on the forecast skill at week 3 (days 19-25). The flow dependent predictability being an inherent feature of the atmosphere-ocean dynamics, affects generally any forecast range and it has been extensively studied. At the extended ranges, when the forecast skill can be generally modest, knowing which flow conditions can lead to a more accurate forecast is particularly relevant. At sub-seasonal and seasonal time scale the higher predictability is typically associated with the distribution of tropical heat anomalies (e.g. ENSO and MJO) or the anomalies in the stratosphere. In this paper the focus is on the forecast skill conditioned by to the flow configuration over the Atlantic sector at initial time. The well-known 4 Euro-Atlantic regimes are used to stratify the initial conditions. The authors show that the forecast initiated with the two phases of NAO are more skillful at week 3 than the forecasts initiated in other flow configuration. This result is consistent with findings from previous work done on flow dependent verification at medium range.

The analysis is limited to the boreal extended winter and it uses 20 years of reforecast data from two sub-seasonal forecast, namely the new version of CNRM and the ECMWF operational system. The skill evaluation is based on the 2m temperature over the Northern Hemisphere. Although the flow dependent skill assessment is based on anomaly correlation, a probabilistic skill estimate is also included.

The study is well presented with an appropriate number of figures and a valuable discussion of the results.

Thank you for your positive review as well as your comments addressed below.

Specific comments:

L 80-85 In the computation of daily forecast anomalies, the model daily climate has been computed as a function of lead-time? Please explain what has been done.

Yes exactly. From each 32-day forecast time series initialized on day D of year Y (For example 01 January 2005), we subtract a 32-day climatology computed as the 19-member ensemble mean, each member being the forecast issued on day D of the 1998-2017 period, except year Y. For the specific case of the CNRM system which is initialized every Thursday, we proceed slightly differently. For example, for a forecast initialized on Thursday 01 January 2005, the 19-member ensemble mean used to build the climatology corresponds to the 19 forecasts issued on the Thursday closest to January 1st each year of the period 1998-2017 (except 2005). This date can thus range from December 29th to January 4th.

Both approaches (CNRM and ECMWF) are synthesized in the manuscript with the index n of the start date for each winter, which is common to both systems (n<=16)

We have slightly rephrased this part to be more specific (see lines 85-87)

L155 It is hard to believe that at week3 and 4 the effect of the initialization shock is still evident.

We assume this comment applies to L.158. When writing this, we were thinking mainly about the initialization of land components (snow cover, soil moisture and deep soil temperature) rather than the atmosphere, for which a potential initial shock will last no longer than a few days. We have rephrased as follows to be more specific:

"The difference could also originate from the better fit between the ECMWF forecast system and the ERA-Interim initial conditions, derived from another version of the same IFS model, in particular for the land surface slow-evolving components such as snow cover, soil moisture and deep soil temperature"

L165 For a "naïve forecast" is intended the climatology? Please explain

"Naive forecast" was not specific enough. We have changed it to "simple climatological forecast"

Comparing Fig.1 and Fig.2 it is clear that the CRPSS is rather more informative than the ACC. In fact, a positive ACC, even if statistically significant, does not translate into a skillful forecast. Why not using the CRPSS as well as ACC for fig.4 and fig.5?

We agree with your comment about the limitations of the correlation to assess the forecast skill. Note that here we make a clear distinction between the grid-point temporal correlation (Fig. 1) based on the 320 forecasts, and the ACC which is a spatial correlation, and can thus apply to individual forecasts.

In our view, the CRPSS is well suited to assess the skill of a prediction system from a large number of forecasts, but less so for the skill of an individual forecast. In this case, the only "probabilistic" information originates from the 10 (or 11) member ensemble, vs. 320 forecasts for the grid point evaluation shown in Fig.2. This is a very reduced sample to compute a robust probabilistic score.

More importantly, the CRPSS is by definition computed over multiple pairs of forecasts and corresponding observations. Here, we aim at assessing how well individual forecasts (individual meaning "one start date") manage to predict the spatial pattern of temperature anomalies. It would be technically feasible to compute the CRPSS for an individual forecast by considering instead the pairs of grid point forecast and their corresponding grid point observation for single events, and then average out over the domain. We consider that this does not make much sense, since it would not inform about the capacity of a single forecast to capture the corresponding observed spatial pattern of temperature anomalies.

Therefore, we would rather pursue with ACCs for individual forecasts in this manuscript.

L205-210 The case of 2009-2010 was characterized by a very persistent negative NAO. Considering that the NAO negative typically has a longer life cycle than the NAO positive (Dawson 2012), a forecast made by persistence could be equally skillful. On the contrary, for the forecast initiated in NAO positive the persistence forecast is expected to be less accurate because the NAO positive life cycle is shorter. By adding an extra bar in Fig.4 indicating the

number of forecasts based on persistence with ACC>0, we could see if the good persistence forecast are actually associated with the NAO negative cases.

We have followed your suggestion (see yellow bars in figure R1 below).

Forecasts based on persistence for week 3 correspond to mean day 1-to-4 forecast temperature anomalies. We find no significant correlation between the NAO index and the yearly number of accurate persistent forecasts. In particular, the 2009-2010 winter does not stand out. This is further discussed in the following comment.

[Figure]

*Figure R1: Same as figure 4 with forecasts based on persistence (yellow bars)*

Results from Fig.6 could indicate that the NAO negative is more persistent in the forecast than the NAO positive. Showing the typical residence time for the NAO positive and for the NAO negative separately using the regimes daily attribution would help to interpret the results. Fig. 9 indicates that for strong NAO conditions the residence time can be 20-30days, could it be that this is mainly associated with the strong NAO negative phase?

We have computed the mean duration (in days) of weather regime sequences of at least 3 days (see Table R1 below). This confirms the fact that the NAO- regime is the most persistent and that persistence times for each weather regime are on average correctly captured by models.

|  | CNRM | ECMWF | ERA5 |
|---|---|---|---|
| NAO+ | 6.1 | 6.1 | 6.1 |
| BLO | 5.5 | 5.7 | 5.7 |
| NAO- | 7.5 | 7.4 | 7.3 |
| AR | 4.9 | 4.9 | 4.9 |

*Table R1: mean persistence time (in days) for WR sequences of 3 days minimum*

However, it could also be that a strong positive (negative) NAO in initial conditions lead to more frequent occurrences of positive (negative) NAO sequences during the forecast. To verify this, we have analyzed for each model the relationship between this number of occurrences and the mean persistence among the forecasts initiated in NAO conditions (see figure R2 below)

[Figure]

*Figure R2: Scatter plot (dots) and associated probability density function (shading) of forecasts initiated in NAO- (top row) and NAO+ (bottom row). The y-axis indicates the number of 3-day or more NAO- (resp. NAO+) sequences in all the ensemble members and the x-axis the mean persistence of these sequences, in days. Red dots mark those forecasts initiated in strong NAO- (resp. NAO+) conditions.*

For both models, the NAO- probability density function has a more elongated shape than the NAO+ counterpart, which is consistent with the well known higher persistence of the NAO- regime. It appears that forecasts with intense NAO- initial conditions (red dots) are overrepresented in the rightmost part of the PDF, thereby confirming to some extent the link between initial intensity and persistence of the NAO- WR. It is not the case for NAO+, although, for ECMWF, many red dots are located on the upper part of the probability density function, meaning that intense initial NAO+ could translate into more frequent occurrences of NAO+ sequences during subsequent weeks. There is no strong evidence for this conclusion given that it does not show in CNRM.

Finally, these additional analyses only provide a very incomplete answer to the main point you have raised about our study. Intense NAO- can lead to increased persistence of this weather regime, which probably depends on the interaction with other drivers (stratosphere ?). On the other hand, the NAO+ case is still unclear, potentially due to the lack of specificity of this weather regime when defined by k-means clustering with k=4 (see our conclusion section as well as our reply to reviewer #2) .

We have decided to report these additional figures and this discussion in supplementary material. This is now indicated in the last sentence before the conclusion section.

Pag.12 For both systems, the forecast initiated with NAO+ indicate higher ACC already by day 5 and beyond day 15 the ACC difference increases.

Following your suggestion and that of Reviewer #2, we have rephrased as follows (line 247):

"For both systems, the mean ACC of the forecasts initialized in NAO+ conditions becomes higher than those initialized with other regimes by day 6 and more markedly from day 15 onwards. albeit not significantly (not shown) "

L 220 A skillful forecast is a forecast that performs better than climatology or than persisting the anomalies of previous 5-8 days. Some of the 68 forecasts that have an ACC>0 in both systems might have a skill close to climatology. Please add some discussion on this.

We agree and have therefore added the following text in section "Reference dataset and forecast skill metrics", lines 115-117

"For the sake of convenience, our definition of skillful individual forecasts is arbitrary, and should be understood as "forecasts with the highest ACCs". It does not imply that they systematically outperform climatological forecasts. This point is addressed by means of the probabilistic skill evaluation (see below)."

L280 By considering the very long period of era5 (1950-2017) we might introduce some important decadal variations. Please comment

As extensively discussed in Hurrell et al (2003), the 1960s were characterized by a higher frequency of NAO- winters, with a sharp reversal in the late 60s and finally predominant NAO+ winters in the early/mid 1990s. This variability does impact the statistics of regime frequencies according to the considered period but not the center of actions themselves. Here, for the composite analysis, we assume a stability of regimes spatial structure in terms of sea-level pressure and temperature anomalies. This stability assumption is somewhat undermined by several studies, such as e.g. Jung et al. (2003) or Woollings et al (2015), and the decadal shift found in the center of actions does impact the intensity and location of temperature and pressure anomaly patterns. However, the changes found do not question the main features of NAO regimes characterized by the Eurasia/Canada temperature dipole and a North Atlantic meridional pressure gradient. The period chosen for our ERA5 composite analysis was therefore a trade-off between a sufficient sample size, requiring more than 20 years, and a stable structure of weather regimes, to be comparable with reforecast composites.

A discussion has been added in lines 295-299

Jung, T., Hilmer, M., Ruprecht, E., Kleppek, S., Gulev, S. K., & Zolina, O. (2003). Characteristics of the recent eastward shift of interannual NAO variability. *Journal of Climate*, *16*(20), 3371-3382.

Woollings, T., Franzke, C., Hodson, D. L. R., Dong, B., Barnes, E. A., Raible, C. C., & Pinto, J. G. (2015). Contrasting interannual and multidecadal NAO variability. *Climate Dynamics*, *45*(1-2), 539-556.

Hurrell, J. W., Kushnir, Y., Ottersen, G., & Visbeck, M. (2003). An overview of the North Atlantic oscillation. *Geophysical Monograph-American Geophysical Union*, *134*, 1-36.

[revised manuscript text omitted]

---

## Author Comment (AC2)

REVIEW of „Flow dependence of wintertime subseasonal prediction skill over Europe" by Ardilouze et al

Review by D. Domeisen

SUMMARY: This study investigates the dependence of winter forecast skill over Europe on the weather regime at initialization of the forecast. The authors find that strong NAO regimes at initialization increase subsequent skill during week 3 in two sub-seasonal prediction models.

ASSESSMENT: The paper is very well written and the analysis is very interesting and should be published. I have a range of minor comments about both the technical and the writing / interpretation aspects of the manuscript, which are detailed below and which I hope will be helpful for the manuscript.

We would like to thank Dr. D. Domeisen for the overall positive comments and also for the thorough and relevant review of our manuscript. We have made our best effort to address each point and hope the revised manuscript is now substantially improved.

My only major comment is that it didn't become fully clear if the skill following intense NAO regimes is mostly a factor of increased persistence due to the intensity of the event or if there is something intrinsically / dynamically different between these regimes. Even if this question cannot fully be answered, it can be addressed in more detail, see comments below.

This point is addressed following your last comment below.

COMMENTS AND RECOMMENDATIONS:

Line 17/18: "predictability well": interesting, I haven't heard this one. Just out of curiosity: I only heard the term "predictability desert" for S2S so far. "well" sounds much more positive, does this refer to a "source" of predictability? So the "windows of opportunity" don't seem like a contradiction, but rather a continuation of the "well"?

This is a very appealing interpretation of our choice to use the word "well" instead of "desert". The truth is we just meant to pick a different word from "desert", originally, without realizing the ambiguity. We thought more of a "Predictability sink" rather than a well but since you mention it, the image of a well fed by intermittent sources of predictability is nicer than a desert or a sink indeed. Identifying and detecting windows of opportunity could indicate when to draw predictability from the well. We wish to keep your interpretation full of imagery and have therefore rephrased as follows: (l.17-24)

'The S2S horizon has been often considered as a "predictability desert" based on mean statistics and traditional methods and analyses inspired from seasonal-to-decadal climate prediction, but the most recent studies reveal instead so-called windows of opportunity based on the fact that under certain circumstances, and for specific events and regions, S2S predictability can be considerably increased (Mariotti et al., 2020). This conditional predictability is illustrated

by a number of case studies showing the successful anticipation of extreme climate events by dynamical forecast systems beyond 15-day lead time (Domeisen et al. in revision).

Rather than a predictability desert, the S2S horizon appears more like a "predictability well' intermittently fed by these windows of opportunity. Timely drawings from the well, i.e. a priori identification of the windows of opportunity, are a major asset in an operational context for the development and uptake of climate services relying on subseasonal forecasts, but it remains a scientific challenge with some promising examples.'

Lines 34 – 37: maybe further citations could help here, e.g. the MJO to NAO teleconnection has been described by Lin et al, 2009 (https://doi.org/10.1175/2008JCLI2515.1), and the resulting S2S predictability has been described in Vitart (2017), which is already cited elsewhere in this manuscript.

Thank you. These two references are now cited.

Line 98/99: "Because ERA5 and ECMWF reforecasts are derived from two versions of the same model": do you mean the use of ERAinterim as initialization for ECMWF as compared to ERA5 for CNRM (table 1)? Please clarify. (in this case, using ERAinterim for validating the results would probably be more suitable than JRA-55, but then again the differences in the results for using ERA5 versus ERAinterim would likely be even smaller than for comparing to JRA-55, so I don't suggest that the authors perform this comparison.)

Actually this sentence was not related to how reforecasts are initialized but rather how they are evaluated. The idea was to avoid any suspicion about ECMWF prediction system being favoured by a reference dataset (ERA5) derived from a similar model. Assessing the skill with JRA-55, arguably independent from the IFS model, allows to discard this suspicion because we do not find any major difference.

We have modified the text as follows (see l. 103-106) to clarify our point:

"To verify if the assessment of the ECMWF predictions is favoured by the choice of this reanalysis, we have compared some of the results obtained with ERA5 to results using the JRA-55 reanalysis as a reference"

Line 112: "the RMSE normalization method is arbitrary": not sure what you mean here, please clarify, which normalization did you use?

It has been normalized by the interquartile range of the observation, as stated in the previous sentence, but here, we just meant that there is no standard way of normalizing the RMSE. We could have normalized it by the observation mean for instance. However, we believe the normalization strategy is of minor importance in the context of this study and hence removed this comment from the revised manuscript. See line 119-120

Section 2.3: do you use a minimum duration of each cluster, or can each consecutive day be assigned to a different weather regime? a persistence criterion may be useful when looking at S2S timescales.

Both have been tested (3-day persistence criterion vs. no persistence criterion, see l. 249-259 of the original manuscript). We have added a few lines at the end of section 2.3 to clarify that both methods have been considered (l.148-150).

Figure 1: it's surprising that all grid points and both models only show significant correlations, with the only exceptions being small areas in weeks 3 and 4. In particular, even correlation values below 0.2 still show significance – is this due to the large number of initializations used here?

Yes, we are positive that even weak correlations appear significant because of the very large sample of 320 initializations for each model. This is suggested by the comments relative to figure 11 a) and b) , where the subsampled initializations lead to patchier correlation patterns. However, we clearly state that significantly positive correlations do not always imply usefulness, hence our CRPSS analysis.

Figure 4: I'm not sure the linear trendlines are very helpful here. Are you suggesting there is a linear trend in forecast skill? For figure 4, I would rather focus this figure on the connection to the NAO only. (I understand you've removed a linear trend, but that is the trend in T2m, not forecast skill or the NAO index, so it's not related to the trend shown in the figure. Good to know removing the trend does not make a difference in your results.)

We thought it was worth assessing the impact of the T2m trend because it could have been that we obtained a larger number of skillful forecasts initialized during the most recent years as compared to forecasts initialized during the early years, for a "bad" reason. For example, we could have expected more skillful forecasts in 2016 than in 1997 just because the recent forecasts tend to predict warm temperature anomalies over Europe more frequently, in agreement with more frequently observed warm anomalies associated to the warming trend.

However, we agree that it is sufficient to indicate that removing the trend does not make a difference. We have thus removed the trend lines from figure 4 and also changed the text accordingly.

Lines 216 – 219: (see also comment above about WR persistence above) I fully agree that a multi-day window should be used.

- Do you allow for several regimes in these 4 days? i.e. could these 4 days theoretically be assigned to 4 different WRs?
- Do you do this analysis separately for each ensemble member, or for the ensemble mean?

Yes, we allow for several regimes in these 4 days. More precisely, we count the occurrence of the 4 regimes for each ensemble member (this now specified in the

manuscript l.233) and each forecast, so that percentages shown in table 2 are based on a pool of 2992 (for ECMWF : 11 members x 4 initial days x 68 skillful forecasts) or 2720 (for CNRM: 10 members x 4 initial days x 68 skillful forecasts) days. We proceed likewise to compute percentages shown in parenthesis, but based on the 252 (=320-68) other forecasts.

- Alternatively, you could introduce a persistence criterion or threshold value for WRs or average the WRs over a few days. We had to introduce such a criterion in this paper http://doi.org/10.5194/wcd-1-373-2020, but I'm sure there are others that do the same. I realize you did this to a certain extent by adding the zero regime in section 3.2, but I'm wondering if the results were more robust overall if you introduced a persistence criterion and a zero category throughout the manuscript.

  The threshold criterion has not been tested in this study but is mentioned at the very end of our conclusion, highlighting the potential limitations of our study stemming from the methodology. As for the persistence criterion we have only tested it for a limited number of analyses (see the dedicated appendix B) but found so little difference that we made the decision to carry out the study without such criterion. Another reason for this decision is that it does not require to discard forecasts characterized by a dominant 'zero' regime at initialization (concerning approx. 70 forecasts out of 320) at the expense of the sample size. Our approach benefits from more robustness due to a larger sample size which is critical for the subsampling approach performed in section 3.4.

  Table 2: do I understand this correctly that each value represents the percentage among the skillful forecasts as opposed to the climatological frequency in brackets? If so, they should add up to 100 (they all do except the skillful forecasts for CNRM, please check).

Thank you for spotting this mistake. Indeed, for CNRM "good" forecasts, the frequency of blocking is 14.9% and not 20.5% (vs. 30.5% for the "wrong" forecasts). The number has been corrected and they now add up to 100%. Fortunately, it does not change the message.

Table 2 caption: "significantly different": I assume you mean that each value is significantly different from the value in brackets? Could you clarify?

Yes, this is correct. We have clarified the caption accordingly

Figure 5: it would be helpful to indicate the number of initializations in brackets next to each WR in the legend

Done

Figure 5: in addition to the significance computed for difference from zero, it would be interesting to know if the ACC is significantly different for NAO+ as compared to other WRs, e.g. by showing error bars or shading (similar to Fig 9) showing the standard deviation to see if NAO+ overlaps with other WRs. I imagine it will not be clearly significantly different, which would not be a problem in my opinion, but it would be nice to get an estimate of the variability of the curves,

e.g. to know if forecasts initialized in NAO+ also contain very poor predictions, or if most of them really show above average ACC. I think this would support the main message of the paper.

We have reproduced figure 5 by adding ±1 standard deviation in dashed thin lines, and extending the y-axis down to -0.5 (fig. R1). As expected, no significant differences can be noticed between ACC curves. The standard deviation intervals are pretty similar even if, by the eye, the NAO- standard deviation interval (light green) seems slightly wider. We choose not to use this plot in the revised paper, but we have rephrased as follows, to also take into account a comment from Reviewer #1:

"For both systems, the mean ACC of the forecasts initialized in NAO+ conditions becomes higher than those initialized with other regimes by day 6 and more markedly from day 15 onwards, albeit not significantly (not shown) " (l.247)

[Figure]

*Figure R1: as Fig.5 with ±1 standard deviations in dashed lines*

Figure 6 / lines 262-263: do you have the same plots for the other two regimes? these would be useful for comparison, as you here make statements about NAO versus non-NAO, but non-NAO initializations are not shown (at least a figure as supplementary material these would be useful). This would also allow for a better understanding if it's the higher persistence of the NAO regimes that makes their aftermath more predictable as compared to other WRs.

The corresponding plots are shown below (fig. R2) and added in supplementary material. Forecasts initiated in BLO (AR) regimes do not show particular changes in BLO (AR) regime proportions at week 3 with respect to climatology. This result may suggest that the higher persistence of NAO regimes play a role in the predictability. But this conclusion remains highly uncertain, as discussed in reply to your last point below.

[Figure]

*Figure R2: Weekly evolution of regime frequency among forecasts initialized in BLO (top row) or AR (bottom row) conditions for CNRM (left column) and ECMWF (right column). The leftmost bar corresponds to the 4 initial days. The rightmost bar corresponds to the climatological frequency for week 3.*

Line 275: "anthropic": do you mean "anthropogenic"? Yes. Corrected

Figure 7 / lines 278 – 279: did I get this correct that all 4 days have to have the same WR for the piControl simulation, while for the S2S data it only has to be the "regime with the greatest

number of occurrence during the 4 initial days" (line 229)? Could you clarify? I understand this will lead to a larger number of samples in piControl, but best to be consistent for comparison.

You understood correctly and we agree that our approach lacks consistency. We have therefore applied the "greatest number of occurrence" method to the piControl and ERA5 as well, and modified the text and composite maps in the revised manuscript accordingly (l. 292-294 and 303). In addition to the better consistency, this looser constraint in the piControl and ERA5 subsampling leads to a larger sample size, which is desirable for the comparison with the S2S data. Thank you for raising this issue.

Figure 7: figure labels would be helpful in addition to the caption. Done

Line 287 – 288: "hemispheric positive AO pattern evoked earlier is a model artefact": this is not clear – I'm pretty sure that all of these patterns will confidently project onto the positive AO pattern, despite their differences.

We have rephrased this assertion, to simply mention the divergence over the North Pacific with respect to the AO+ loading pattern (l. 305-306).

Line 300: "This agreement is much better for negative than positive NAO": I'm wondering if this is due to the fact that NAO- is a much more pronounced North Atlantic regime than NAO+. In particular, if dividing up WRs into more than 4 regimes, NAO- remains a separate regime (equal to Greenland blocking), while NAO+ is sub-divided into separate regimes by the clustering algorithm. NAO+ is more of a mixture of several regimes that reflect the average state of the North Atlantic, while NAO- is a distinct regime. To paraphrase Brian Hoskins (I hope I'm doing this correctly), NAO+ is basically the "normal" state of the North Atlantic, while NAO- is a distinct anomalous state of the North Atlantic.

In the study carried out by Falkena et al (2020) and cited in our conclusion, they suggest 6 WR instead of 4, with NAO+ still present, although with a slight shift in the centres of action. It is no longer the dominant WR though. However, their study meets your comment when they assert in their conclusion:

"the dominant occurrence of the NAO+ when there are only four clusters, which likely is due to it being the only regime with a low pressure area in the north, is reduced by the addition of two regimes that also have this feature. Therefore, six regimes allow for more variability in their representation of the circulation and prevent all data with a more zonal flow from projecting onto the NAO+"

Another interesting approach is that of Dorrington and Strommen (2020) . They show that removing the influence of the jet speed leads to an optimal clustering of k=3 or k=5 without NAO+, and with an increased stability of the remaining regimes

We have added a sentence suggesting that  our clustering method may not be optimal, with NAO+ being a mere generic mode, and the other regimes are perturbations from this generic

mode. Another interpretation is that NAO+ is not specific enough and potentially conceals a variety of WR. See line 319-320 and lines 410-414 in the conclusion.

Dorrington, J., & Strommen, K. J. (2020). Jet speed variability obscures Euro‑Atlantic regime structure. *Geophysical Research Letters*, *47*(15), e2020GL087907.

Falkena, S. K., de Wiljes, J., Weisheimer, A., & Shepherd, T. G. (2020). Revisiting the identification of wintertime atmospheric circulation regimes in the Euro‑Atlantic sector. *Quarterly Journal of the Royal Meteorological Society*, *146*(731), 2801-2814.

Line 311 – 315: this decorrelation timescale and behavior (e.g. the rebound) is consistent when looking at the decorrelation for a wide range of different NAO indices (Figure 3b in http://doi.org/10.1175/JCLI-D-17-0226.1, already cited elsewhere in this article).

True. A comment has been added (lines 334-335)

Lines 331 – 332 / lines 370 onward: I don't think that the regression analysis is proof that the NAO pattern at initialization influences the entire NH. There are many common remote drivers that will lead to both a NAO-type pattern over the North Atlantic and consistent anomalies elsewhere, e.g. precursors in the tropics, the North Pacific, or the stratosphere. If you want to include Figure 10 (it would be equally fine in supplementary), I think the text should be formulated more carefully. This might also explain your finding on lines 343-344: "However, no improvement of skill is detected over South East US and off the US Atlantic coast, as could have been expected from the teleconnection patterns."

We agree that this is an overstatement. We rephrased the sentences at the beginning of section 3.4 as follows (lines 351-353):

"In the previous section, we have identified a statistical link between wintertime temperature anomalies over a number of regions of the northern hemisphere extratropics and the 3-week antecedent NAO index."

And later in the same section (lines 362-366) as:

"These regions match remarkably well with the regression patterns highlighted in observations (fig. 10) and they are relatively consistent between CNRM and ECMWF systems. Note that these regression patterns do not necessarily imply a causal relationship and may originate from a number of remote drivers (which are not investigated here). Indeed, no improvement of skill is detected over South East US and off the US Atlantic coast."

Line 335: earlier only the top 10% of strong NAO initializations were kept. What is the reason for using the quartile now? Increased sample size?

The idea was to keep a sample large enough (40 forecasts out of 320) to draw robust conclusions, but it is a bit arbitrary. We have added a sentence to justify this choice (l. 357)

Line 354: "but performs reasonably well anyway": can you be more specific / quantitative?

We have changed the text as follows:

" but the spatial patterns compare relatively well". Spatial correlations between CNRM and ECMWF skill is discussed in Section 3.1

Section 3.4 / Figure 6: It didn't become fully clear if the skill following intense NAO regimes is mostly a factor of increased persistence due to the initial intensity of the event or if there is something intrinsically / dynamically different between these regimes (see major comment above). I think it might help to repeat Figure 6 for initializations in intense NAO regimes and to check if a clearer pattern emerges as compared to all "regular" NAO regimes and other WRs.

 This is a key point indeed, and we have repeated figure 6 as suggested (see fig. R3 below). We do find that these forecasts show an increased proportion of NAO- ( NAO+ to a lesser extent) at week 3, which may indicate that the persistence of these regimes is related to their initial intensity.

[Figure]

[Figure]

*Figure R3: same as figure 6 after subsampling forecasts with the top 10% of strong NAO+ (top row) or NAO- (bottom row) initializations.*

However, we wanted to go a bit further, and see if it is more the regimes persistence or their number of occurrences that depended on the initial intensity.

The scatterplots of figure 4 show for each forecast initiated in NAO- (NAO+) the number of occurrences of NAO- (NAO+) sequences of at least 3 consecutive days, as a function of the mean duration of these sequences. The red dots correspond to the forecasts with the highest initial NAO intensity.

For both models, the NAO- probability density function has a more elongated shape than the NAO+ counterpart, which is consistent with the well known higher persistence of the NAO- regime. It appears that forecasts with intense NAO- initial conditions (red dots) are overrepresented in the rightmost part of the PDF, thereby confirming to some extent the link between initial intensity and persistence of the NAO- WR. It is not the case for NAO+, although, for ECMWF, many red dots are located on the upper part of the probability density function, meaning that intense initial NAO+ could translate into more frequent occurrences of NAO+ sequences during subsequent weeks. There is no strong evidence for this conclusion given that it does not show in CNRM.

Finally, these additional analyses only provide a very incomplete answer to the main point you have raised about our study. An intense NAO- can lead to increased persistence of this weather regime, which probably depends on the interaction with other drivers (stratosphere ?). On the other hand, the NAO+ case is still unclear, potentially due to the lack of specificity of this weather regime when defined by k-means clustering with k=4 (see discussion of a previous point) .

We have decided to report these additional figures and this discussion in supplementary material. This is now indicated before the conclusion section (l. 371-376).

[Figure]

*Figure R4: Scatter plot (dots) and associated probability density function (shading) of forecasts initiated in NAO- (top row) and NAO+ (bottom row). The y-axis indicates the number of 3-day or more NAO- (resp. NAO+) sequences in all the ensemble members and the x-axis the mean persistence of these sequences, in days. Red dots mark those forecasts initiated in strong NAO- (resp. NAO+) conditions.*

SOME TYPOS I FOUND:

Thanks for reporting the typos. Corrected typos are marked with a red "check" symbol

Lines 9 and 367: conditionned -> conditioned ✔

Line 11: others parts -> other parts ✔

Line 18: traditionnal -> traditional ✔

Line 19: reveals -> reveal ✔

Line 68: of **the** new CNRM system ✔

line 92: greenhouse gases emissions -> greenhouse gas emissions ✔

line 175: tend -> tends ✔

line 192: than other forecasts more spread out -> than other forecasts **that are** more spread out✔

line 193: figured in green and yellow shades -> **plotted** in green and yellow shades**. (missing period)** ✔

Line 194: 2x counterparts ✔

Line 206: forecast -> forecasts ✔

Line 214: next section -> the next section ✔

Table 2 caption: these frequency -> these frequencies ✔

Line 287: over North Pacific -> over the North Pacific ✔

Line 287: similitude -> similarity ✔

Line 307: the figure 9 -> figure 9 ✔

Line 363: regime -> regimes ✔

Figure A1 caption: "for week 1 to week for": do you mean "week 4"? Yes ☺ ✔

[revised manuscript text omitted]